# HRGS: Hierarchical Gaussian Splatting for Memory-Efficient High-Resolution 3D Reconstruction

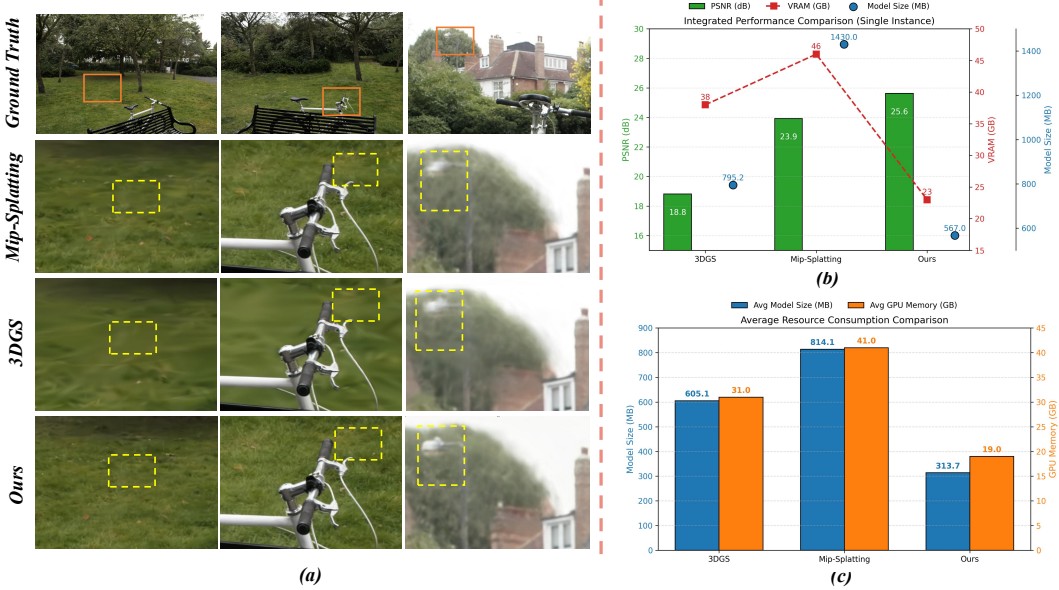

Figure 1: (a) High-resolution ($\sim$ 5K) renderings of the "bicycle" scene from the Mip-NeRF 360 dataset, with results from 3DGS, Mip-Splatting, and our method. Red dashed boxes highlight key details. (b) Performance on this scene: our method achieves the highest PSNR (25.6 dB) with significantly lower GPU memory (23 GB) and model size (567 MB) than 3DGS and Mip-Splatting. (c) Average resource usage across the full dataset shows our method maintains the smallest memory and model footprint.

## Abstract

3D Gaussian Splatting (3DGS) has achieved significant progress in real-time 3D scene reconstruction. However, its application in high-resolution reconstruction scenarios faces severe memory scalability bottlenecks. To address this issue, we propose Hierarchical Gaussian Splatting (HRGS), a memory-efficient framework with hierarchical block-level optimization from coarse to fine. Specifically, we first derive a global, coarse Gaussian representation from low-resolution data; we then partition the scene into multiple blocks and refine each block using high-resolution data. Scene partitioning comprises two steps: Gaussian partitioning and training data partitioning. In Gaussian partitioning, we contract irregular scenes into a normalized, bounded cubic space and employ a uniform grid to evenly distribute computational tasks among blocks; in training data partitioning, we retain only those observations that lie within their corresponding blocks or make significant contributions to the rendering results. By guiding each block's refinement with the global coarse Gaussian prior, we ensure alignment and seamless fusion of Gaussians across adjacent blocks. To reduce computational resource demands, we introduce an Importance-Driven Gaussian Pruning (IDGP) strategy: during each block's refinement, we compute an importance score for every Gaussian primitive and remove those with minimal rendering contribution, thereby accelerat-

ing convergence and reducing redundant computation and memory overhead. To further enhance surface reconstruction quality, we also incorporate normal priors from a pretrained model. Finally, even under memory-constrained conditions, our method enables high-quality, high-resolution 3D scene reconstruction. Extensive experiments on three public benchmarks demonstrate that our approach achieves state-of-the-art performance in high-resolution novel view synthesis (NVS) and surface reconstruction tasks.

# 1 INTRODUCTION

3D scence reconstruction remains a longstanding challenge in computer vision and graphics. A significant advancement in this domain is the Neural Radiance Field (NeRF) (Mildenhall et al., 2021), which effectively represents geometry and view-dependent appearance using multi-layer perceptrons (MLPs), demonstrating significant advancements in 3D reconstruction quality. Recently, 3D Gaussian Splatting (3DGS) (Kerbl et al., 2023) has gained considerable attention as a compelling alternative to MLP-based (Mildenhall et al., 2021) and feature grid-based representations (Chen et al., 2022; Fridovich-Keil et al., 2022; Liu et al., 2020; Müller et al., 2022c). 3DGS stands out for its impressive results in 3D scene reconstruction and novel view synthesis while achieving real-time rendering at 1K resolutions. This efficiency and effectiveness, combined with the potential integration into the standard GPU rasterization pipeline, marks a significant step toward the practical adoption of 3D reconstruction methods. Although 3DGS has demonstrated impressive 3D reconstruction results, its application in high-resolution scenarios encounters critical memory scalability limitations. Specifically,when reconstructing outdoor scenes at ultra-high resolutions approaching 5K (e.g. $4978 \times 3300$ pixels) in standardized benchmark datasets like Mip-NeRF 360 (Barron et al., 2022a), conventional 3DGS implementations demand excessive VRAM, exceeding the capacity of mainstream GPUs with limited memory, such as the NVIDIA A5000 (24GB VRAM). This computational bottleneck arises from the increasing resolution: higher resolutions demand more GPU memory, as illustrated in Fig. 1. Such algorithmic behavior fundamentally conflicts with finite GPU memory resources, resulting in catastrophic memory overflow during optimization phases.

To overcome these critical memory constraints while preserving reconstruction fidelity for high-resolution scene reconstruction, we present **Hierarchically Gaussian Splatting (HRGS)**, a memory-efficient framework with hierarchical block optimization from coarse to fine. Specifically, we first obtain a coarse global Gaussian representation using low-resolution images. Subsequently, to minimize memory usage on a single GPU, we partition the scene into spatially adjacent blocks and parallelly refined. Each block is represented with fewer Gaussians and trained on reduced data, allowing further optimization with high-resolution images. The partitioning strategy operates at two levels: Gaussian primitives and training data. To achieve a more balanced partition of Gaussians and avoid blocks with sparse Gaussians, we begin by contracting unbounded Gaussians. In detail, we define a bounded cubic region and use its boundary to normalize the Gaussian positions. Within this region, Gaussians are contracted via a linear mapping, while those outside undergo nonlinear contraction, yielding a more compact Gaussian representation. We then apply a uniform grid subdivision strategy to this contracted space, ensuring an even distribution of computational tasks. During data partitioning for training, we compute the SSIM loss (Wang et al., 2003) for each observation by comparing two renderings: One rendering is implemented with the complete global Gaussian representation, while the other is executed subsequent to the elimination of Gaussians within the target block. A more pronounced SSIM loss denotes that the observation exerts a more substantial contribution to the target block, so we set a threshold on SSIM loss and retain only observations whose values exceed it. To mitigate artifacts at the block boundaries, we further include observations that fall within the region of the considered block. Finally, to prevent overfitting, we employ a binary search algorithm during data partitioning to expand each block until the number of Gaussians it contains exceeds a specified threshold. This innovative strategy effectively reduces interference from irrelevant data while improving fidelity with decreased memory usage, as demonstrated in Tab. 4.

After partitioning the Gaussian primitives and data, we initialize each block in the original, uncontracted space using the coarse global Gaussian representation. To accelerate convergence and reduce computational overhead during block-level refinement with high-resolution data, we introduce an Importance-Driven Gaussian Pruning (IDGP) strategy. Specifically, we evaluate the interaction between each Gaussian and the multi-view training rays within the corresponding block, and discard

those with negligible rendering contributions. All blocks are then refined in parallel, and subsequently integrated into a unified, high-resolution global Gaussian representation. To further enhance the quality of the reconstructed surfaces, we incorporate the View-Consistent Depth-Normal Regularizer (Chen et al., 2024a), which is applied both during the initialization of the coarse global Gaussian representation and throughout the subsequent block-level refinement. Finally, our method enables high-quality and high-resolution scene reconstruction even under constrained memory capacities (e.g., NVIDIA A5000 with 24GB VRAM). We validate our method on two sub-tasks of 3D reconstruction: high-resolution NVS and surface reconstruction, and demonstrate that it delivers superior high-resolution reconstruction performance. In summary, the main contributions of this paper are:

- We propose HRGS, a memory-efficient coarse to fine framework that leverages low-resolution global Gaussians to guide high-resolution local Gaussians refinement, enabling high-resolution scene reconstruction with limited GPU memory.

- We introduce a novel partitioning strategy for Gaussian primitives and data, optimizing memory usage, reducing irrelevant data interference, and enhancing reconstruction fidelity.

- We propose a novel dynamic pruning strategy, Importance-Driven Gaussian Pruning (IDGP), which evaluates the contribution of each Gaussian primitive during training and selectively removes those with low impact. This approach significantly improves training efficiency and optimizes memory utilization.

- Extensive experiments on three public datasets demonstrate that our approach achieves state-of-the-art performance in high-resolution rendering and surface reconstruction.

## 2 RELATED WORK

**3D Reconstruction.** Recent 3D reconstruction research can be broadly categorized into traditional geometry-based and deep learning methods. The former relies on multi-view stereo (MVS) (YAN, 2021) and structure from motion (SfM) (Schonberger & Frahm, 2016) to estimate scene depth and camera poses, producing point clouds and subsequent surface meshes. The latter integrates implicit functions (e.g., SDF, Occupancy) (Huang et al., 2023) with volumetric rendering for high-fidelity reconstruction, as exemplified by Neural Radiance Fields (NeRF) (Mildenhall et al., 2021). However, NeRF-based approaches often struggle with real-time performance in large-scale or dynamic scenarios. In contrast, 3D Gaussian Splatting (Kerbl et al., 2023) encodes scenes as 3D Gaussians (with position, scale, and color), using differentiable point-based rendering to achieve fast training and inference while balancing accuracy and quality. Balancing high fidelity, scalability, and real-time capability remains a key challenge in 3D reconstruction. Within the field of 3D reconstruction, there are primarily two main sub-tasks: novel view synthesis (NVS) and surface reconstruction.

**Novel View Synthesis.** Novel View Synthesis (NVS) aims to generate a target image from an arbitrary camera pose, given source images and their camera poses (Levoy & Hanrahan, 1996; Gortler et al., 1996). NeRF (Mildenhall et al., 2021) integrates implicit representations with volume rendering(Drebin et al., 1988; Levoy, 1990), demonstrating impressive results in view synthesis. However, dense point sampling remains a major bottleneck for rendering speed. To address this, various methods accelerate NeRF by replacing the original multi-layer perceptrons (MLPs) (Chen & Zhang, 2019; Park et al., 2019) with discretized representations, such as voxel grids (Sun et al.), hash encodings (Müller et al., 2022a), or tensor radiation fields (Chen et al., 2022). Additionally, some approaches (Yariv et al., 2023; Reiser et al., 2023) distill pretrained NeRFs into sparse representations, enabling real-time rendering. Recent advancements in 3D Gaussian Splatting (3DGS) have significantly improved real-time rendering, demonstrating that continuous representations are not strictly necessary. However, directly optimizing and rendering at high resolutions drastically increase memory overhead, making it challenging to achieve real-time reconstruction of high-quality scenes on mainstream GPUs with limited memory (24GB). Our approach specifically addresses this challenge by reducing the computational cost of high-resolution processing while preserving reconstruction fidelity.

**Multi-View Surface Reconstruction.** Traditional multi-view stereo methods (Bleyer et al., 2011a; Broadhurst et al., 2001; Kutulakos & Seitz, 2000; Schönberger et al., 2016a; Seitz & Dyer, 1999;

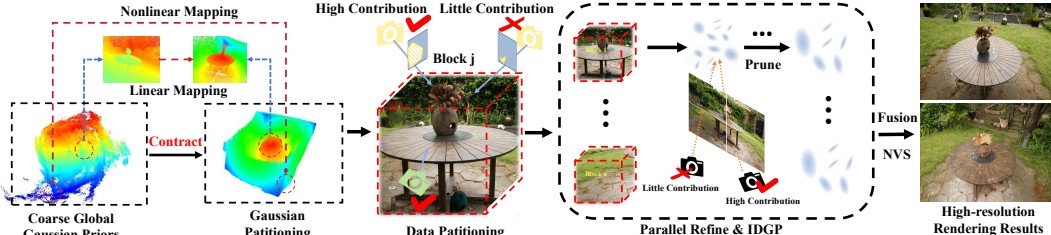

Figure 2: **Illustrative diagram of the hierarchical block optimization framework.** We first derive a global coarse Gaussian representation using low-resolution data, which is then contracted into a bounded cubic region. Subsequently, the contracted Gaussian primitives are partitioned into blocks, each paired with corresponding data. Leveraging the global coarse Gaussian as initialization, we parallelly refine each block in the original uncontracted space using high-resolution data. During this refinement process, an Importance-Driven Gaussian Pruning strategy is employed to compute the interaction between each Gaussian primitive and training view rays, removing low-contribution primitives to accelerate convergence and reduce redundancy. The optimized blocks are then concatenated to form the final global Gaussian representation, which is validated through novel view synthesis (NVS) and surface reconstruction tasks.

Seitz et al., 2006) reconstruct scenes by estimating dense depth maps (Bleyer et al., 2011b; Schönberger et al., 2016b), fusing them into point clouds (Furukawa & Ponce, 2010; Lhuillier & Quan, 2005), and generating surfaces through triangulation or implicit fitting (Kazhdan & Hoppe, 2023). While widely adopted, these approaches often suffer from artifacts, noise, and local minima during reconstruction (Barnes et al., 2009). Recent advances in neural implicit representations, such as NeRF (Mildenhall et al., 2022) and SDF-based variants (Wang et al., 2021; Yu et al., 2022), learn continuous volumetric or surface fields directly from images, jointly modeling geometry and appearance for improved robustness to occlusions and textureless regions. However, their high computational cost and limited scalability remain challenges. To address these issues, 3D Gaussian Splatting (3DGS) (Kerbl et al., 2023) uses explicit anisotropic Gaussians for efficient, differentiable rasterization (Yifan et al., 2019). However, its performance in sparse or large-scale settings is limited by insufficient geometric supervision (Chen et al., 2023). Recent methods, like VCR-GauS (Chen et al., 2024b), Vastgaussian (Lin et al., 2024), and SuGaR (Guédon & Lepetit, 2024), improve reconstruction through view-consistent constraints (Turkulainen et al., 2024; Bae & Davison, 2024b; Yin et al., 2019). Our HRGS framework advances 3DGS by combining global Gaussian priors, adaptive partitioning, and importance-driven pruning, supporting high-fidelity 5K rendering under strict memory constraints. This makes HRGS a promising solution for high-resolution surface reconstruction, overcoming limitations in previous methods (Li et al., 2024; Barron et al., 2022c; Fang & Wang, 2024).

## 3 METHODOLOGY

Our proposed HRGS efficiently reconstructs high-resolution scenes. We first review 3DGS in Section 3.1. Next, in Section 3.2, we present the memory-efficient coarse-to-fine framework, detailing the partitioning of Gaussian primitives and data, along with the proposed Importance-Driven Gaussian Pruning (IDGP) strategy. Finally, Section 3.3 describes the loss function employed in our approach.

### 3.1 PRELIMINARY

We begin with a brief overview of 3D Gaussian Splatting (3DGS) (Kerbl et al., 2023). In the 3DGS framework, a scene is represented as a set of discrete 3D Gaussian primitives, denoted by $G_K = \{G_k \mid k = 1, \ldots, K\}$, where $K$ is the total number of Gaussians in the scene. Each Gaussian $G_k$ is defined by a set of learnable parameters, including its 3D position $\mathbf{p}_k \in \mathbb{R}^{3 \times 1}$, opacity $\sigma_k \in [0, 1]$, and geometric properties, which typically consist of scaling and rotation parameters that define the Gaussian covariance matrix $\Sigma_k \in \mathbb{R}^{3 \times 3}$. Furthermore, spherical harmonic (SH) features $f_k \in \mathbb{R}^{3 \times 16}$ are used to encode view-dependent color information $c_k \in \mathbb{R}^{3 \times 1}$, allowing for a realistic depiction of color variations as a function of the viewing angle. For rendering purposes,

the combined color and opacity contributions from multiple Gaussians at a given pixel are weighted according to their respective opacities. The color blending for overlapping Gaussians is computed as follows:

$$\hat{C} = \sum_{k \in M} c_k \alpha_k \prod_{j=1}^{k-1} \left(1 - \alpha_j\right), \tag{1}$$

where $c_k$ and $\alpha_k = \sigma_k G_k$ denote the color and density of the $k$-th Gaussian primitive, respectively.

### 3.2 HIERARCHICAL BLOCK OPTIMIZATION FRAMEWORK

Traditional 3D Gaussian methods (Kerbl et al., 2023; Chen et al., 2024a) rely on global iterative optimization for scene reconstruction but struggle with memory inefficiency in high-resolution settings, such as the Mip-NeRF 360 (Barron et al., 2022b) dataset. To address this, we propose a hierarchical optimization framework that balances coarse global representation and fine-grained local refinement, as shown in Fig 2. We first construct a low-resolution global Gaussian prior, guiding block-wise high-resolution optimization to enhance geometric detail while maintaining memory efficiency. This approach enables precise reconstruction under constrained memory conditions. The following subsections detail the coarse global Gaussian generation, Gaussian and data partitioning strategies, as well as refinement and post-processing procedures.

**Coarse Global Gaussian Representation.** This stage establishes the foundation for subsequent Gaussian and data partitioning. Initially, we train the COLMAP (Schönberger et al., 2016; Schönberger & Frahm, 2016) points using all observations at a low resolution for 30,000 iterations, generating a coarse representation of the global geometric structure. The resulting Gaussian primitives are represented as $G_K = \{G_k \mid k = 1, \ldots, K\}$, where $K$ denotes the total number of Gaussians. In the following block-wise high-resolution refinement, this robust global geometric prior ensures that Gaussians are positioned accurately, thereby preventing drift and eliminating inter-block discontinuities, minimizing significant fusion artifacts.

**Primitives and Data Division.** Directly applying uniform grid division in the original 3D space may lead to uneven Gaussian distribution in local regions (e.g. many nearly empty grid cells alongside overly dense ones). To address this imbalance, we define a bounded cubic region and contract all Gaussians within it. Within this region, the central one-third of the space is designated as the internal region, while the surrounding area is classified as the external region. The internal region is bounded by the minimum and maximum corner positions, $\mathbf{p}_{\min}$ and $\mathbf{p}_{\max}$, which define the limits of the central one-third of the entire region. To standardize the representation of global Gaussians, we introduce a normalization step: $\hat{\mathbf{p}}_k = 2\left(\mathbf{p}_k - \mathbf{p}_{\min}\right)/\left(\mathbf{p}_{\max} - \mathbf{p}_{\min}\right) - 1$. As a result, the coordinates of Gaussians located in the internal region are constrained within the range $[-1, 1]$. To achieve more effective contraction of the global Gaussians, we apply a linear mapping for the Gaussians in the internal region, while a nonlinear mapping is employed for the external region (as shown in Fig. 2). The final contraction step is performed using the function described in (Wu et al., 2023):

$$\text{contract}(\hat{\mathbf{p}}_k) = \begin{cases} \hat{\mathbf{p}}_k, & \text{if } \|\hat{\mathbf{p}}_k\|_\infty \leq 1, \\ \left(2 - \frac{1}{\|\hat{\mathbf{p}}_k\|_\infty}\right) \frac{\hat{\mathbf{p}}_k}{\|\hat{\mathbf{p}}_k\|_\infty}, & \text{if } \|\hat{\mathbf{p}}_k\|_\infty > 1. \end{cases} \tag{2}$$

The contracted space is then uniformly partitioned into $n$ blocks (the specific number of blocks used will be discussed further in Sec. 4.), resulting in a more balanced Gaussian partitioning. After partitioning the Gaussians, our objective is to ensure that each block is sufficiently trained. In other words, the training data assigned to each block should be highly relevant to the region it represents, focusing on refining the details within the block. To achieve this, we select observations and retain only those that contribute significantly to the visible content of the corresponding block in the rendering results. Since SSIM loss effectively captures structural differences and is somewhat robust to brightness variations (Wang et al., 2003), we use it as the foundation for our data partition strategy. Specifically, for the $j$-th block, the global Gaussians contained within it are represented as: $G_{Kj} = \{G_k \mid b_{j,\min} \leq \text{contract}(\hat{\mathbf{p}}_k) < b_{j,\max}, k = 1, \ldots, K_j\}$, where $b_{j,\min}$ and $b_{j,\max}$ define the spatial bounds of the $j$-th block, and $K_j$ is the number of Gaussians contained within the block. The set of observations assigned to the $j$-th block is defined by the following formula:

$$\mathbf{P}_j^1 = \text{Mask}\left(\mathcal{L}_{\text{SSIM}}\left(I_{G_K}(\boldsymbol{\tau}), I_{G_K \setminus G_{Kj}}(\boldsymbol{\tau})\right) > \epsilon\right) \odot \boldsymbol{\tau}, \tag{3}$$

where $\text{Mask}(\cdot)$ generates an element-wise binary mask. Each element of the mask is set to 1 if it satisfies the condition inside the mask (i.e. the SSIM loss exceeds a threshold $\epsilon$), and 0 otherwise. The term $G_K \setminus G_{Kj}$ denotes the portion of the global set $G_K$ excluding the block $G_{Kj}$. $\boldsymbol{\tau}$ is a matrix containing all camera poses, with each column $\tau_i$ representing the $i$-th camera pose. $\odot$ is element-wise product operation. And the resulting

set $\mathbf{P}_j^1$ represents the camera poses assigned to the $j$-th block. However, this strategy does not account for the projection of the considered block, which may lead to artifacts at the edges of the block. To address this issue, we further include poses that fall within the boundaries of the considered block:

$$\mathbf{P}_j^2 = \text{Mask}\left(b_{j,\min} \leq \text{contract}(\hat{\mathbf{p}}_{\tau_i}) < b_{j,\max}\right) \odot \boldsymbol{\tau}. \tag{4}$$

where $\hat{\mathbf{p}}_{\tau_i}$ is the position under the world coordinate of pose $i$. The final assignment is:

$$\mathbf{P}_j(\boldsymbol{\tau}, G_{Kj}) = \text{Merge}\left(\mathbf{P}_j^1, \mathbf{P}_j^2\right), \tag{5}$$

where $\text{Merge}$ denotes the concatenate operator that removes any duplicate elements, ensuring only one copy of each element is retained. To prevent overfitting, we employ a binary search method (Lin, 2019) to incrementally expand $b_{j,\min}$ and $b_{j,\max}$ until $K_j$ exceeds a predefined threshold. Notably, this procedure is applied exclusively during the data partitioning phase for each block.

**Importance-Driven Gaussian Pruning (IDGP).** After the Gaussian primitives and data division, we proceed to train each block in parallel in the original uncontracted space. Specifically, we first initialize each block using the coarse global Gaussian prior, and then fine-tune each block using high-resolution data as detailed in Sec. 3.3. During block-level optimization, we further accelerate convergence and reduce redundancy by applying a lightweight importance scoring and pruning strategy. Let $\mathcal{R}_b$ denote the set of all rays cast from the training views assigned to block $b$. For each Gaussian primitive $p_i$ in block $b$, we only consider its interactions with $\mathcal{R}_b$ and define the weighted hit count as

$$H_i = \sum_{r \in \mathcal{R}_b} \mathbf{1}(p_i \cap r)\, T_{i,r}, \text{ where } T_{i,r} = \prod_{\substack{p_k \cap r \\ \text{depth}(p_k) < \text{depth}(p_i)}} \left(1 - \alpha_k\right). \tag{6}$$

Here, $\mathbf{1}(p_i \cap r) = 1$ if and only if ray $r$ intersects $p_i$, and $T_{i,r}$ accumulates the transmission up to $p_i$ by all closer primitives $p_k$. We then compute the raw volume of $p_i$ as $v_i = \prod_{d=1}^{3} s_{i,d}$, where each $s_{i,d}$ is the scale factor of $p_i$ along the $d$-th spatial axis, and apply logarithmic compression $\widetilde{v}_i = \ln(1 + v_i)$. Finally, we assign each primitive an importance score with its opacity $\alpha_i$: $S_i = \alpha_i\, \widetilde{v}_i\, H_i$. After evaluating $\{S_i\}$ for all primitives in the block, we sort them in descending order and remove the lowest 20%. The remaining Gaussians, now both globally informed by the coarse prior and locally pruned of low-impact points, continue through block-level fine-tuning. Finally, we select the fine-tuned Gaussians within each block and, guided by the global geometric prior, concatenate the blocks to obtain the fine-tuned global Gaussian. Through this process, the previously coarse global Gaussians are significantly enhanced in areas where they lacked detail.

### 3.3 Loss Function

To optimize both the coarse and refined stages, the loss functions are defined as follows. First, we use the RGB loss $\mathcal{L}_{RGB}$ from 3DGS for the novel view synthesis task. To reconstruct scene surfaces, we enforce normal priors $\mathbf{N}$ predicted by a pretrained monocular deep neural network (Bae & Davison, 2024a) to supervise the rendered normal map $\hat{\mathbf{N}}$ using **L1** and cosine losses:

$$\mathcal{L}_n = \|\hat{\mathbf{N}} - \mathbf{N}\|_1 + (1 - \hat{\mathbf{N}} \cdot \mathbf{N}). \tag{7}$$

Additionally, to effectively update Gaussian positions, we utilize the predicted normal $\mathbf{N}$ from the pretrained model to supervise the D-Normal $\overline{\mathbf{N}}_d$. The D-Normal is derived from the rendered depth by computing the cross-product of horizontal and vertical finite differences from neighboring points:

$$\overline{\mathbf{N}}_d = \frac{\nabla_v \mathbf{d} \times \nabla_h \mathbf{d}}{|\nabla_v \mathbf{d} \times \nabla_h \mathbf{d}|}, \tag{8}$$

where $\mathbf{d}$ represents the 3D coordinates of a pixel obtained via back-projection from the depth map. We then apply the D-Normal regularization from (Chen et al., 2024a):

$$\mathcal{L}_{dn} = w \cdot \left(\|\bar{\mathbf{N}}_d - \mathbf{N}\|_1 + (1 - \bar{\mathbf{N}}_d \cdot \mathbf{N})\right), \tag{9}$$

where $w$ is a confidence term. The overall loss function integrates these components:

$$\mathcal{L}_{total} = \mathcal{L}_{RGB} + \lambda_1 \mathcal{L}_s + \lambda_2 \mathcal{L}_n + \lambda_3 \mathcal{L}_{dn}, \tag{10}$$

where $\lambda_1$, $\lambda_2$, and $\lambda_3$ balance the individual terms. The term $\mathcal{L}_s$ is introduced to simplify depth computation, as described in (Chen et al., 2024a).

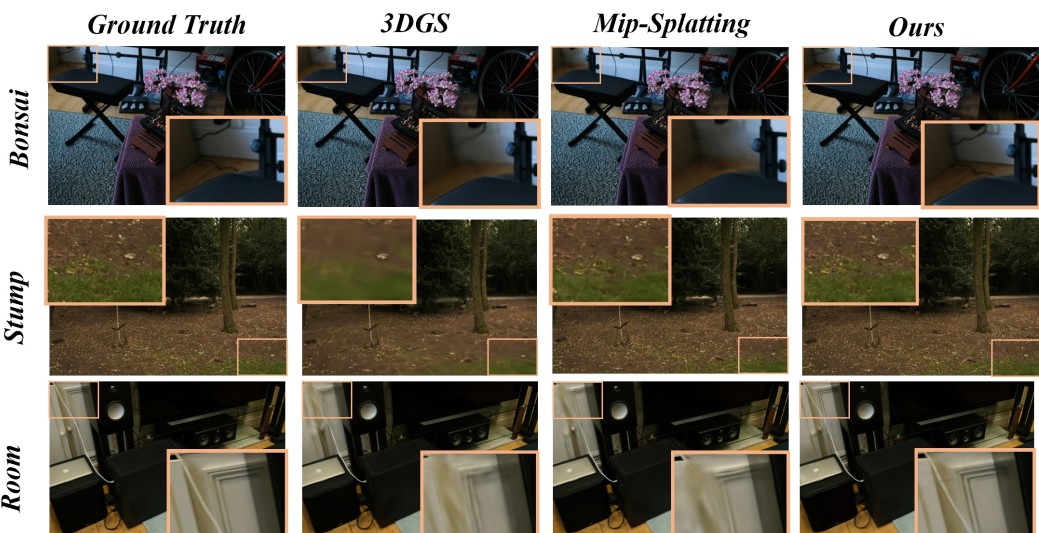

Figure 3: **Qualitative Comparison on the Mip-NeRF 360 Dataset.** Three representative scenes demonstrate that our method more faithfully preserves fine-scale structures and achieves superior visual fidelity compared to 3DGS and Mip-Splatting.

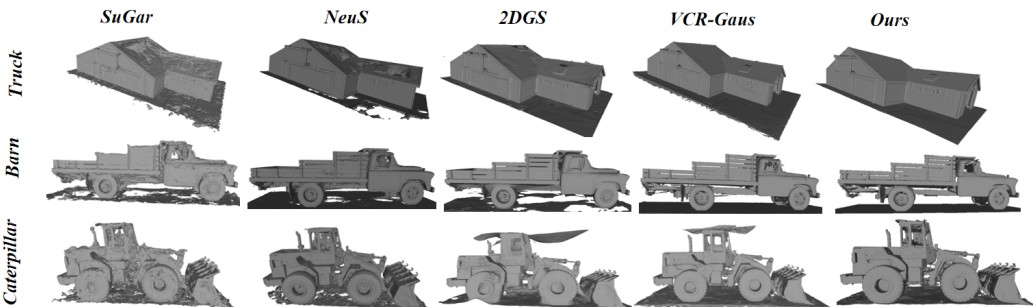

Figure 4: **Qualitative Comparison on TNT dataset.** Reconstructions from left to right—SuGar, NeuS, 2DGS, and VCR-Gaus—demonstrate that our method delivers more complete surface geometry, enhanced smoothness in planar regions, and superior preservation of fine structural details, thereby outperforming existing approaches in geometric fidelity.

# 4 EXPERIMENTS

## 4.1 EXPERIMENTAL SETUPS

**Dataset and Metrics.** To evaluate the effectiveness of our reconstruction method, we conduct experiments on two core tasks: novel view synthesis (NVS) and surface reconstruction, using multiple benchmark datasets. Our primary goal is high-resolution 3D reconstruction under constrained memory, which emphasizes the preservation of fine details (e.g., textures, edges); we therefore select datasets offering high-resolution imagery and rich geometric complexity. We first assess high-resolution NVS performance on Mip-NeRF360 (Barron et al., 2022b) (which includes scenes at resolutions such as 4946×3286), followed by high-fidelity surface reconstruction on the Tanks and Temples (TNT)(Knapitsch et al., 2017) dataset. Additionally, we perform comparative experiments on the Replica(Straub et al., 2019) dataset to further validate our method. For a comprehensive evaluation, we employ standard metrics including SSIM, PSNR, LPIPS, and F1-score. Rendering efficiency is also assessed in terms of frames per second (FPS).

**Implementation Details.** We begin by following the 3DGS (Kerbl et al., 2023) pipeline, performing 30,000 iterations at a low resolution (0.3K) to obtain a coarse global Gaussian prior. During this stage, we introduce our Importance-Driven Gaussian Pruning (IDGP) strategy, which scores the rendering contribution of each Gaussian primitive and prunes those with the lowest impact. This step prevents irrelevant viewpoints from being assigned to training blocks in subsequent stages, reducing unnecessary computational overhead. The resulting coarse prior serves as initialization for the refinement phase. In the contraction stage, we define

Table 1: **Mip-NeRF 360 Full-Resolution Results.** The rendering quality comparison highlights the best and second-best results.

|  | Zip-NeRF | Instant-NGP | Mip-NeRF | 3DGS | 3DGS+EWA | Mip-Splatting | **Ours** |
|---|---|---|---|---|---|---|---|
| PSNR ↑ | 28.25 | 24.36 | 27.51 | 26.19 | 26.42 | 26.53 | 28.41 |
| SSIM ↑ | 0.822 | 0.644 | 0.779 | 0.795 | 0.783 | 0.833 | 0.869 |
| LPIPS ↓ | 0.198 | 0.366 | 0.254 | 0.349 | 0.347 | 0.343 | 0.245 |

Table 2: **Quantitative Results on the Tanks and Temples Dataset (Knapitsch et al., 2017).** The best results are highlighted in **orange**, while the second-best results are marked in **blue**.

|  | NeuS-based | | | Gaussian-based | | | |
|---|---|---|---|---|---|---|---|
| **Scene** | NeuS | MonoSDF | SuGaR | 3DGS | 2DGS | VCR-GauS | Ours |
| Barn | 0.29 | 0.49 | 0.14 | 0.13 | 0.36 | 0.62 | 0.65 |
| Caterpillar | 0.29 | 0.31 | 0.16 | 0.08 | 0.23 | 0.26 | 0.29 |
| Courthouse | 0.17 | 0.12 | 0.08 | 0.09 | 0.13 | 0.19 | 0.23 |
| Ignatius | 0.83 | 0.78 | 0.33 | 0.04 | 0.44 | 0.61 | 0.64 |
| Meetingroom | 0.24 | 0.23 | 0.15 | 0.16 | 0.16 | 0.19 | 0.24 |
| Truck | 0.45 | 0.42 | 0.26 | 0.18 | 0.16 | 0.52 | 0.61 |
| Mean | 0.38 | 0.39 | 0.19 | 0.09 | 0.30 | 0.40 | 0.45 |
| FPS | <10 | | | 159 | 68 | 145 | 146 |

the central one-third of the full scene as the internal region and the remainder as the external region. The contracted Gaussians are then divided into four spatial sub-blocks. For data assignment, we use an SSIM threshold of $\epsilon = 0.1$. Each sub-block is further trained for 30,000 iterations. Specifically, we apply IDGP at the 10,000th, 15,000th, and 25,000th iterations to prune low-impact Gaussians based on their interaction contributions with training rays. This dynamic pruning accelerates convergence and reduces computational redundancy. To facilitate surface reconstruction, we adopt the depth-normal regularization method described in Sec. 3.3. Specifically, we use the pretrained DSINE (Bae & Davison, 2024a) model for outdoor scenes and the pretrained GeoWizard (Fu et al., 2024) for indoor scenes to predict normal maps. The hyperparameters $\lambda_1, \lambda_2$, and $\lambda_3$ are set to 1, 0.01, and 0.015, respectively. After rendering the depth maps, we perform truncated signed distance function (TSDF) fusion and process the results using Open3D (Zhou et al., 2018). Additional details are provided in the supplementary.

**Novel View Synthesis.** As shown in Tab. 1, we compare our method with several existing approaches, including mip-NeRF (Barron et al., 2022a), Instant-NGP (Müller et al., 2022b), zip-NeRF (Barron et al., 2023), 3DGS (Kerbl et al., 2023), 3DGS+EWA (Zwicker et al., 2001), and Mip-Splatting (Yu et al., 2024). At high resolutions, our method significantly outperforms all state-of-the-art techniques. As shown in Fig. 3, our method produces high-fidelity imagery devoid of fine-scale texture distortions. While 3DGS (Kerbl et al., 2023) introduces noticeable erosion artifacts due to dilation operations, Mip-Splatting (Yu et al., 2024) shows improved performance, yet still exhibits evident texture distortions. In contrast, our method avoids such issues, producing images that are both aesthetically pleasing and closely aligned with the ground truth, demonstrating the effectiveness of our hierarchicall refined strategy.

Table 3: **Our experimental comparsions on Replica (Straub et al., 2019). Bold** indicates the best.

| Type | Method | F1-score | Time |
|---|---|---|---|
| Implicit | NeuS | 65.12 | >10h |
|  | MonoSDF | **81.64** |  |
| Explicit | 3DGS | 50.79 | ≤2h |
|  | SuGar | 63.20 |  |
|  | 2DGS | 64.36 |  |
|  | Ours | **74.87** |  |

**Surface Reconstruction.** Our method not only delivers high-quality novel view synthesis but also enables accurate 3D surface reconstruction. As shown in Tab. 2, our approach outperforms both NeuS-based methods (e.g., NeuS (Wang et al., 2021), MonoSDF (Yu et al., 2022), and Geo-NeuS (Fu et al., 2022)) and Gaussian-based techniques (e.g., 3DGS (Kerbl et al., 2023), SuGaR (Guédon & Lepetit, 2024), 2DGS (Huang et al., 2024a), and VCR-GauS (Chen et al., 2024b)) on the Tanks and Temples (TNT) dataset. Compared to NeuS-based approaches, our method achieves significantly faster reconstruction. Compared to Gaussian-based methods, our method obtains substantially better reconstruction quality, for instance, improving the F1-score from 0.3 to 0.45 compared to 2DGS. Moreover, our approach surpasses the recent state-of-the-art method VCR-GauS, achieving a higher reconstruction quality (0.45 vs. 0.4). As illustrated in Fig. 4, our method excels at recovering fine geometric details. We also observe a significant advantage in rendering speed, outperforming 2DGS by more than a factor of two. On the Replica dataset, as summarized in Tab. 3, our method attains performance comparable to MonoSDF (Yu et al., 2022) while operating at substantially higher speeds. Further-

more, compared to explicit reconstruction approaches including 3DGS (Kerbl et al., 2023), SuGaR (Guédon & Lepetit, 2024), and 2DGS (Huang et al., 2024a), our method achieves notably higher F1-scores.

## 4.2 ABLATION STUDIES

To validate the effectiveness of individual components in our method, we conducted a series of ablation experiments on the "Stump" scene from the Mip-NeRF 360 dataset and the "Ignatius" scene from the TNT dataset. Specifically, we evaluated the impacts of the following components: hierarchical block optimization strategy, Importance-Driven Gaussian Pruning (IDGP), and data partitioning strategy.

Table 4: **Ablation on Data Division.** "SO Ass." refers to SSIM-based assignment, while "BO Ass." denotes boundary-based assignment. **Bold** indicates the best.

| Method | Settings | | | | |
|---|---|---|---|---|---|
| | baseline | w/o contraction | w/o SO Ass. | w/o BO Ass. | Full |
| PSNR ↑ | 21.87 | 22.73 | 24.29 | 22.96 | **25.24** |
| F1 ↑ | 0.55 | 0.54 | 0.58 | 0.60 | **0.64** |

**Ablation of the Data Division.** As shown in Tab. 4, we analyzed the impact of the data partitioning strategy, using the original Gaussian global prior as the baseline. The results in the first and last columns of Tab. 4 demonstrate the effectiveness of our proposed method in improving performance (0.55 vs. 0.64). The second column in Tab. 4 further indicates that assigning relevant data in the contracted space is essential for enhancing reconstruction quality. The third column in Tab. 4 highlights the importance of Strategy 1 (Eq. (13)) in data partitioning, and we also found that Strategy 2 (Eq. (14)) plays a significant role in preventing artifacts at the edges of blocks. ß

**Ablation of the Number of Blocks.** As shown in Tab. 5, we investigate how the number of blocks affects reconstruction performance by splitting the coarse global Gaussian into 2, 4, 8, or 16 blocks. Our results indicate that too few blocks can cause conflicts between local and global optima, resulting in insufficient refinement of fine details, whereas too many blocks may lead to imbalanced data distribution and local overfitting. Consequently, we select four blocks for our experiments on the TNT dataset.

Table 5: **Ablation on number of blocks.**

| | 2 | 4 | 8 | 16 |
|---|---|---|---|---|
| **PSNR** ↑ | 24.86 | **25.24** | 23.93 | 22.56 |
| **F1** ↑ | 0.62 | **0.64** | 0.61 | 0.57 |

Table 6: **Ablation Studies on the "Stump" Scene of the Mip-NeRF 360 Dataset (Barron et al., 2022a).**

| Method | Model Size(MB) | GPU Memory(G) | PSNR | SSIM | LPIPS |
|---|---|---|---|---|---|
| Baseline | 348.62 | 23 | 26.39 | 0.804 | 0.291 |
| Baseline w/o IDPG | 601.04 | 28 | 26.41 | 0.791 | 0.288 |

**Ablation of Importance-Driven Gaussian Pruning.** To validate the effectiveness of the proposed Importance-Driven Gaussian Pruning (IDGP) strategy, we conducted an ablation study on the "stump" scene of the Mip-NeRF 360 dataset. As shown in Tab. 6, we compare the full method with the IDGP mechanism (Baseline) against a control variant in which IDGP is disabled (Baseline w/o IDGP). IDGP is able to selectively prune redundant Gaussians during training without degrading rendering quality, thereby achieving significant improvements in both model structure and computational efficiency.

## 5 CONCLUSIONS

In this work, we propose HRGS, a memory-efficient coarse-to-fine framework that uses low-resolution global Gaussians to guide the refinement of high-resolution local Gaussians, enabling high-resolution scene reconstruction under limited GPU memory. Our novel partitioning strategy for Gaussian primitives and data effectively facilitates block-wise optimization, significantly alleviating the high memory overhead typical of traditional 3DGS in high-resolution 3D reconstruction. Despite reduced memory requirements, our method achieves superior reconstruction quality and demonstrates state-of-the-art performance on two key sub-tasks of 3D reconstruction. As a result, this work establishes a new baseline for high-resolution 3D reconstruction, setting an important precedent for future research in the field.

# Ethics Statement

This study proposes a high-resolution 3D reconstruction method based on Hierarchical Random Gaussian Splattering (HRGS), aiming to achieve high-quality scene reconstruction under limited memory conditions. This work falls under the category of basic algorithm research. All experiments are conducted using publicly available standard benchmark datasets (such as Mip-NeRF 360, Tanks and Temples, and Replica). The use of these datasets complies with academic conventions and does not involve human subjects, personal data, or any form of privacy risks. We encourage the application of this technology in fields that benefit society, including education, digitalization of cultural heritage, smart city visualization, and virtual/augmented reality. We also call on users to abide by relevant ethical guidelines and laws and regulations. Although this method has advantages in improving reconstruction efficiency, it may also be misused in unauthorized scene reconstruction and other behaviors that infringe on privacy. Therefore, we suggest that in practical deployment, enhanced ethical review and legal supervision should be implemented to ensure the legitimacy of the application of this technology. The authors declare no potential conflicts of interest.

# Reproducibility Statement

To ensure the reproducibility of this study, we have provided a detailed description of the HRGS framework in the main text and the Method section (Section 3), including the hierarchical block optimization strategy, Gaussian and data partitioning methods, and the Importance-Driven Gaussian Pruning (IDGP) strategy. The Experimental Setup section (Section 4) clearly specifies the datasets used, evaluation metrics, hardware environment (NVIDIA A5000/A800 GPU), and software configuration (PyTorch 2.0.1, CUDA 11.7). We will make the complete code implementation publicly available on GitHub after the final revision of the paper, including model initialization, partitioning process, training scripts, and loss function definitions, to enable other researchers to reproduce our results. In addition, Appendix A further provides hyperparameter settings, resolution scaling experiments, and module ablation analysis, which enhances the transparency and verifiability of the method. If necessary, we are willing to provide training logs, model weights, and preprocessed data to support the community's further verification and development of this work.

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

# A APPENDIX

| *Ground Truth* | *3DGS* | *Mip-Splatting* | *Ours* |

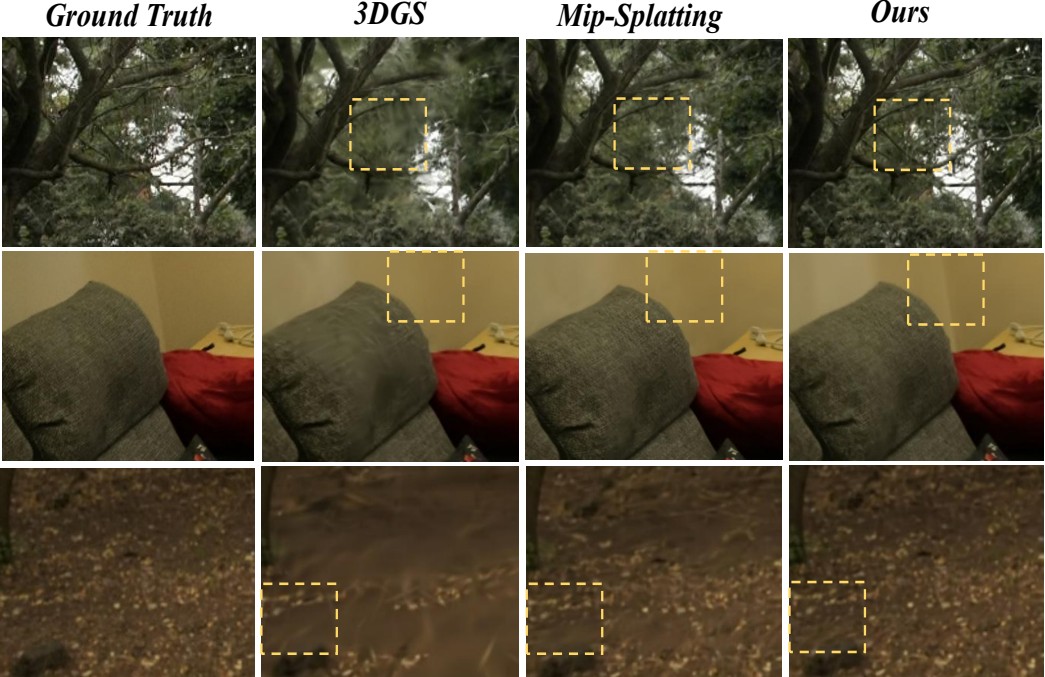

Figure A: Comparison of 3DGS, Mip-Splatting and Ours at full resolution on the Mip-NeRF 360 dataset.

## A.1 IMPLEMENTATION DETAILS

All experiments are conducted using machines equipped with NVIDIA A5000 GPUs, with PyTorch 2.0.1 and CUDA 11.7 as the software environment. For comparison methods that exceed the memory capacity of the A5000 setup, we employ NVIDIA A800 GPUs to ensure reliable execution and fair evaluation. Unless otherwise specified, we adopt the same hyperparameter settings as 3DGS (Kerbl et al., 2023). For outdoor scenes in the TNT dataset (Knapitsch et al., 2017), we incorporate decoupled appearance modeling (Lin et al., 2024) to mitigate exposure-related artifacts. All models are trained and evaluated on the same data splits as used in 2DGS (Huang et al., 2024a), across the TNT (Knapitsch et al., 2017) and Mip-NeRF360 (Barron et al., 2022a) benchmarks.

## A.2 ADDITIONAL EFFICIENCY COMPARISONS

We have supplemented the efficiency metrics in the table A. HRGS exhibits clear advantages in both memory consumption and the number of active Gaussians. Although the training time is longer, this reflects a deliberate time-for-space trade-off, enabling high-resolution reconstruction under strict memory constraints and demonstrating efficiency in resource-limited scenarios.

Table A: Efficiency comparison of different methods at full resolution on the 360 dataset.

| Method | Time (h) | GPU (GB) | Gaussians num |
|---|---|---|---|
| 3DGS | 3 | 31 | 2,754,212 |
| Mip-Splatting | 2.5 | 41 | 4,013,411 |
| Ours | 5 | 19 | 1,301,166 |

## A.3 ADDITIONAL ABLATION STUDY

In this section, we further investigate the contributions of our proposed modules through comprehensive ablation studies. We begin by analyzing the block-wise training strategy in Section A.3.1. Next, we examine

Table B: **Mip-NeRF 360 Full-Resolution Results. Bold** indicates the best.

|  | Zip-NeRF | Instant-NGP | Mip-NeRF | 3DGS | CityGS | Mip-Splatting | **Ours** |
|---|---|---|---|---|---|---|---|
| PSNR ↑ | 28.25 | 24.36 | 27.51 | 26.19 | 27.71 | 26.53 | **28.41** |
| SSIM ↑ | 0.822 | 0.644 | 0.779 | 0.795 | 0.801 | 0.833 | **0.869** |
| LPIPS ↓ | **0.198** | 0.366 | 0.254 | 0.349 | 0.294 | 0.343 | 0.241 |

the impact of the Data Division and IDGP strategies in Section A.3.2 and Section A.3.3, respectively. While these ablation studies were previously presented in the main paper, they were limited to a single scene. Here, we extend the evaluation to the full dataset, providing a more comprehensive and statistically robust analysis. Additionally, we introduce a ablation study on normal priors in Section A.3.4 to evaluate the effectiveness of geometric constraints. Finally, in Section A.3.5, we assess the scalability and effectiveness of our method across various resolution scales, demonstrating its consistent performance under different computational constraints.

### A.3.1 BLOCK-WISE TRAINING FOR HIGH-RESOLUTION RECONSTRUCTION

As shown in Tab. B, our method achieves superior rendering quality compared to CityGS. As illustrated in Fig. B, our approach demonstrates exceptional detail preservation in intricate geometric features and textural nuances. Additionally, Tab. C highlights the advantages of our block partitioning strategy over CityGS in enhancing model scalability. Across the Mip-NeRF 360 dataset, our method consistently reduces the total model size. For example, in the bicycle scene, the model size is 587.05 MB compared to the baseline's 1126.4 MB, with comparable improvements observed at the block level Tab. C. Furthermore, as shown in Fig. C, our method substantially reduces GPU memory consumption compared to 3DGS, Mip-Splatting, and CityGS, underscoring its superior optimization of computational efficiency over CityGS despite employing similar block partitioning strategies.

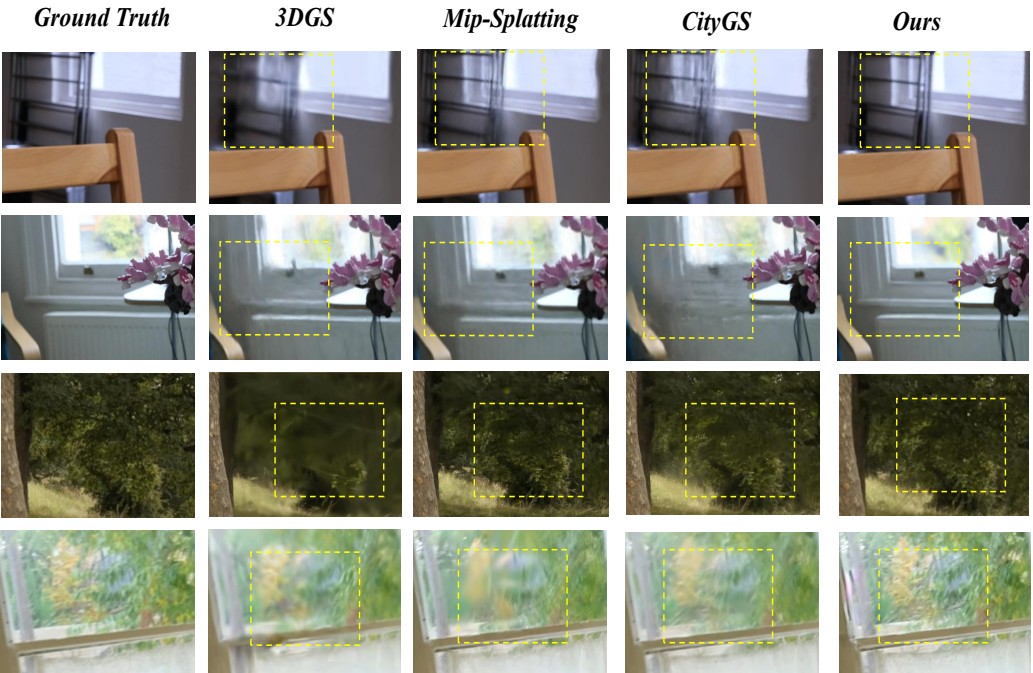

| *Ground Truth* | *3DGS* | *Mip-Splatting* | *CityGS* | *Ours* |

Figure B: **Comparison of 3DGS, Mip-Splatting**, CityGS and Ours at full resolution on the Mip-NeRF 360 dataset.

### A.3.2 ABLATION OF THE DATA DIVISION.

As shown in Table. D, we conducted chunk-based ablation experiments on the entire TNT dataset to evaluate the impact of data-partitioning strategies, using the original Gaussian global prior as the baseline. We system-

Table C: **Comparison of model sizes (MB) across CityGS and Ours. Bold** indicates the best.

| Scene | CityGS (MB) | | | | | Ours (MB) | | | | |
|---|---|---|---|---|---|---|---|---|---|---|
| | Total | cell0 | cell1 | cell2 | cell3 | Total | cell0 | cell1 | cell2 | cell3 |
| bicycle | 1126.4 | 512.63 | 480.45 | 70.92 | 63.11 | **587.05** | 187.22 | 238.32 | 100.76 | 61.75 |
| bonsai | 278.52 | 49.97 | 43.83 | 117.54 | 67.18 | **160.10** | 49.81 | 47.16 | 48.18 | 34.01 |
| counter | 215.85 | 21.38 | 22.26 | 78.73 | 93.49 | **144.58** | 23.06 | 36.63 | 36.36 | 45.54 |
| flowers | 738.18 | 269.72 | 276.41 | 96.18 | 93.87 | **399.71** | 163.86 | 133.96 | 65.55 | 63.33 |
| garden | 1196.80 | 269.72 | 276.41 | 96.18 | 93.87 | **360.52** | 114.76 | 86.78 | 87.60 | 79.39 |
| kitchen | 324.08 | 30.26 | 26.46 | 157.15 | 110.21 | **170.94** | 38.41 | 35.78 | 76.55 | 47.52 |
| stump | 873.41 | 399.37 | 302.45 | 171.59 | 191.51 | **368.62** | 103.86 | 103.32 | 80.75 | 80.69 |
| treehill | 745.24 | 273.96 | 236.07 | 136.24 | 98.97 | **385.84** | 109.68 | 149.32 | 66.55 | 60.29 |
| room | 301.02 | 33.58 | 94.33 | 74.08 | 106.03 | **221.97** | 316.31 | 33.84 | 75.64 | 76.19 |

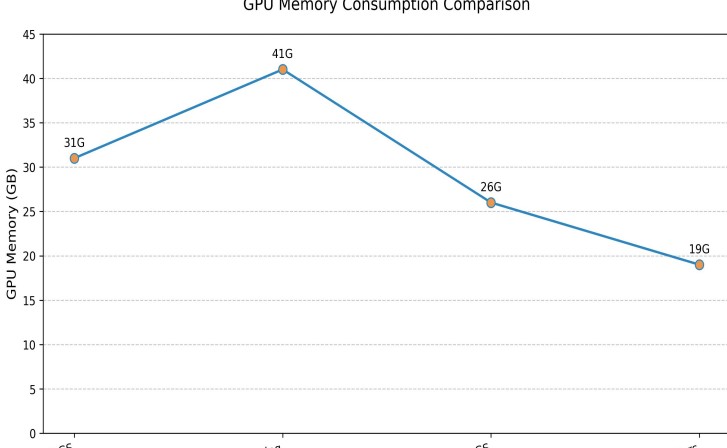

Figure C: **Average GPU memory consumption of 3DGS, Mip-Splatting, CityGS, and our method on the Full-Resolution Mip-NeRF 360 dataset.**

atically investigate the contribution of individual components, including contraction, SSIM-based assignment (SO Ass.) and boundary-based assignment (BO Ass.).

The results in the first and last columns of Tab. D demonstrate the effectiveness of our proposed method in improving performance (0.36 vs. 0.45). The second column in Tab. D further indicates that assigning relevant data in the contracted space is essential for enhancing reconstruction quality. The third column in Tab. D highlights the importance of SO Ass. in data partitioning, and we also found that BO Ass. plays a significant role in preventing artifacts at the edges of blocks.

Table D: **Ablation on Data Division (performed on the full TNT dataset).** "SO Ass." refers to SSIM-based assignment, while "BO Ass." denotes boundary-based assignment. **Bold** indicates the best.

| Method | Settings | | | | |
|---|---|---|---|---|---|
| | baseline | w/o contraction | w/o SO Ass. | w/o BO Ass. | Full |
| PSNR ↑ | 22.19 | 23.63 | 24.89 | 23.96 | **25.02** |
| F1 ↑ | 0.36 | 0.38 | 0.41 | 0.43 | **0.45** |

### A.3.3 ABLATION OF IMPORTANCE-DRIVEN GAUSSIAN PRUNING.

To evaluate the impact of Importance-Driven Gaussian Pruning (IDGP), we performed ablation studies on the full-resolution Mip-NeRF 360 dataset by comparing the proposed method with a variant that excludes IDGP. As shown in Tab. F, removing IDGP results in a substantial increase in model size from 313.72 MB to 621.04 MB, along with a 42% rise in GPU memory usage (19 GB to 27 GB). Although PSNR slightly improves (28.02 vs. 27.91), the SSIM drops notably (0.821 vs. 0.863). These results underscore the effectiveness of IDGP in eliminating redundant Gaussians, significantly enhancing model efficiency while preserving high-quality recon-

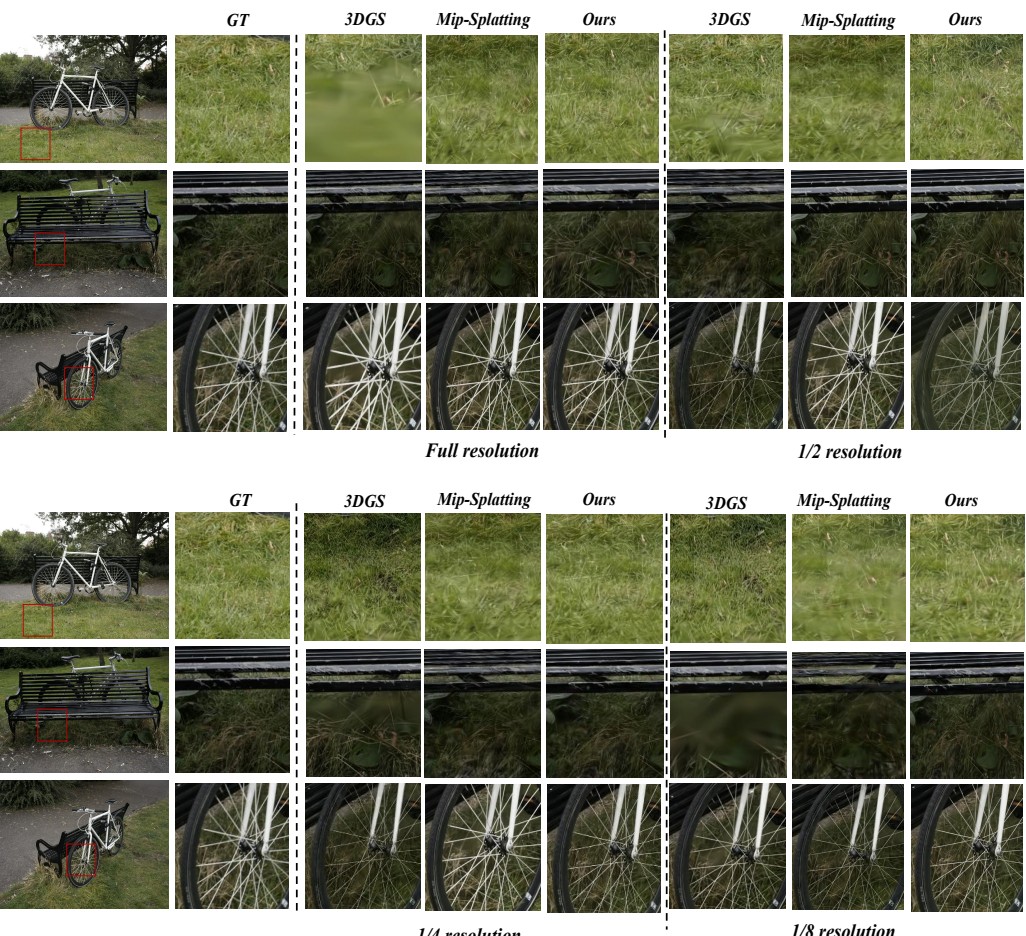

Figure D: **Qualitative comparsion on Mip-NeRF 360 for different resolution scales.**

struction, thus highlighting its importance for memory-efficient 3D reconstruction in high-resolution settings. To evaluate the impact of our hierarchical refinement strategy, we conducted an ablation study on the Mip-NeRF 360 dataset. As illustrated in Tab. H, the results demonstrate that employing only coarse 3D Gaussian Splatting with pruning yields limited reconstruction quality. In contrast, our complete pipeline—integrating both coarse initialization and hierarchical refinement—achieves a substantial improvement, confirming the effectiveness of the proposed refinement stages.

### A.3.4 CONTRIBUTION OF PRETRAINED GEOMETRIC PRIORS

Regarding this study,our technical solution introduces normals estimated by the Dsine normal predictor, applying them as normal loss supervision to the normals generated by the depth2normal module in the 2DGS framework. We conducted experiments on the TNT dataset, and the results **??** indicate that this supervision strategy improves the F1 score by 0.04. While it promotes reconstruction performance to a certain extent, the improvement effect is relatively limited.

As shown in the table E, our method achieves superior normal estimation quality while maintaining competitive computational efficiency compared to the baseline 2DGS approach.

Table E: Quantitative comparison of normal estimation performance.

| Method | 2DGS | 2DGS + Dsine | Ours |
|---|---|---|---|
| F1 Score | 0.30 | 0.34 | 0.45 |

Table F: **Ablation study on IDPG across the entire full resolution Mip-NeRF 360 Dataset(Barron et al., 2022a).**

| Method | Model Size(MB) | GPU Memory(G) | PSNR | SSIM | LPIPS |
|---|---|---|---|---|---|
| w/o IDPG | 621.04 | 27 | 28.02 | 0.821 | 0.308 |
| Ours (Full) | 313.72 | 19 | 27.91 | 0.863 | 0.342 |

### A.3.5 ROBUSTNESS VALIDATION ACROSS RESOLUTION SCALES

To thoroughly evaluate the robustness of our method across varying resolution scales, we performed detailed visual comparisons of rendering quality on the Mip-NeRF 360 dataset at multiple resolution levels. As shown in Tab. G, we compare our method with several existing approaches, including mip-NeRF (Barron et al., 2022a), Instant-NGP (Müller et al., 2022b), zip-NeRF (Barron et al., 2023), 3DGS (Kerbl et al., 2023), 3DGS+EWA (Zwicker et al., 2001), and Mip-Splatting (Yu et al., 2024). Our approach demonstrates comparable performance to these prior methods at one-eighth of the original resolution. Furthermore, at higher resolutions, our method significantly outperforms all state-of-the-art techniques. As shown in Fig. D, our method produces high-fidelity imagery devoid of fine-scale texture distortions. While 3DGS (Kerbl et al., 2023) introduces noticeable erosion artifacts due to dilation operations, Mip-Splatting (Yu et al., 2024) shows improved performance, yet still exhibits evident texture distortions. In contrast, our method avoids such issues, producing images that are both aesthetically pleasing and closely aligned with the ground truth, demonstrating the effectiveness of our hierarchicall refined strategy.

Table G: **Quantitative results on Mip-NeRF 360 (Downscaled Resolutions).** The best results are highlighted in **orange**, while the second-best results are marked in **blue**.

| Method | 1/2x | | | 1/4x | | | 1/8x | | |
|---|---|---|---|---|---|---|---|---|---|
| | PSNR↑ | SSIM↑ | LPIPS↓ | PSNR↑ | SSIM↑ | LPIPS↓ | PSNR↑ | SSIM↑ | LPIPS↓ |
| zip-NeRF | 30.00 | 0.892 | 0.099 | 31.57 | 0.933 | 0.056 | 32.52 | 0.954 | 0.037 |
| Instant-NGP | 25.23 | 0.719 | 0.251 | 26.54 | 0.900 | 0.142 | 28.42 | 0.877 | 0.092 |
| mip-NeRF | 29.19 | 0.864 | 0.136 | 30.45 | 0.912 | 0.077 | 30.86 | 0.931 | 0.058 |
| 3DGS | 26.75 | 0.783 | 0.274 | 27.31 | 0.823 | 0.181 | 29.19 | 0.880 | 0.107 |
| Mip-Splatting | 26.47 | 0.801 | 0.305 | 27.66 | 0.823 | 0.181 | 29.39 | 0.884 | 0.108 |
| Ours | 31.03 | 0.902 | 0.168 | 31.02 | 0.921 | 0.138 | 31.24 | 0.903 | 0.096 |

Table H: Ablation study on coarse global 3DGS with IDGP at full resolution on the 360 dataset.

| Method | PSNR | SSIM | LPIPS |
|---|---|---|---|
| Coarse 3GDS + IDGP | 19.60 | 0.617 | 0.472 |
| Full pipeline | **28.41** | **0.869** | **0.241** |

### A.4 MESH EXTRACTION

The mesh extraction method used is consistent with 2DGS (Huang et al., 2024b). Given rendered depth maps and camera poses, these inputs are fused via Open3D's TSDF integration to construct a continuous Signed Distance Field (SDF). The final surface mesh is then directly extracted from the SDF at its zero-level isosurface using Marching Cubes. enabling direct geometry reconstruction without intermediate point cloud representations. Additionally, no post-processing is applied to the final mesh.

### A.5 ANTI-ALIASING 3D RECONSTRUCTION

The robustness of our method (HRGS) across multiple resolutions is primarily attributed to its hierarchical coarse-to-fine design integrated with a global prior. Initially, the model learns a global coarse Gaussian representation from low-resolution data, constructing a stable and resolution-independent structural scaffold of the scene. This global prior serves as an anchor for subsequent high-resolution optimization. Consequently, even with variations in training resolution, the refinement process consistently begins from this unified geometric foundation, thereby preventing significant deviations and maintaining the integrity of the overall scene structure.

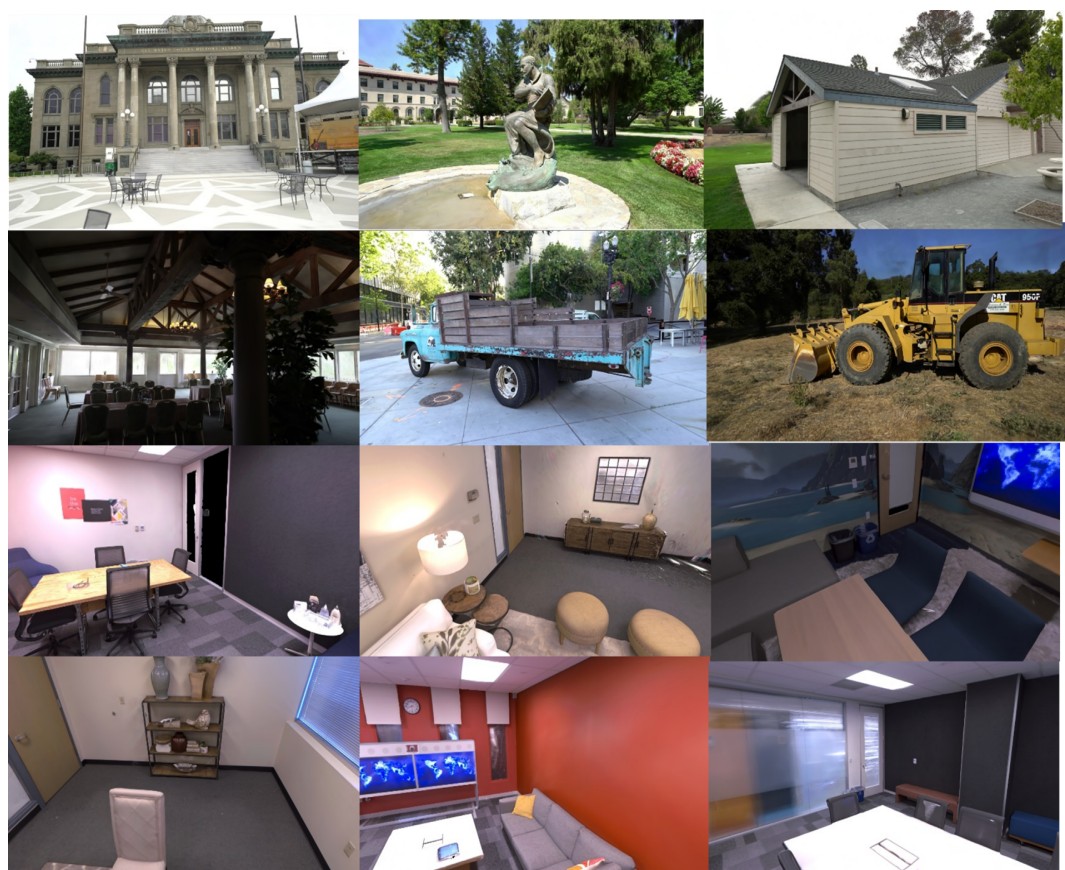

Figure E: **Qualitative rendering results on TNT and Replica dataset.**

Regarding the mitigation of aliasing artifacts (e.g., erosion), HRGS explicitly decouples global structure modeling from local detail refinement through a block-wise optimization strategy guided by data selection. Each local block refines details based on the stable coarse model, which provides essential global contextual information. This approach effectively suppresses jagged edges and erosion artifacts that typically emerge during direct high-resolution training. Essentially, the coarse global stage establishes an anti-aliasing prior, while the block-wise refinement enhances spatial details without overfitting to noisy or redundant patterns.

### A.6  ADDITIONAL QUALITATIVE RESULTS

As shown in Fig.A, our method preserves fine-scale structures more accurately and achieves higher visual fidelity than 3DGS(Kerbl et al., 2023) and Mip-Splatting (Yu et al., 2024) across three representative scenes. Fig. F displays the rendering (top) and surface reconstruction (bottom) results on the Mip-NeRF360 (Barron et al., 2022a) dataset. Additional rendering results on the TNT (Knapitsch et al., 2017) and Replica (Straub et al., 2019) datasets are provided in Fig. E. Collectively, these visual comparisons substantiate our method's capability for high-quality 3D reconstruction while maintaining critical geometric details.

### A.7  LIMITATIONS

Although our method can achieve high-resolution scene reconstruction under limited GPU resources, extending the block-based framework to dynamic scenes introduces new challenges. In particular, ensuring temporal consistency across frames and accurately modeling motion across spatial partitions remain open problems. Addressing these challenges will be a key focus of our future work, with the goal of enabling temporally coherent and spatially consistent reconstruction in dynamic environments.

*Render*

*Mesh*

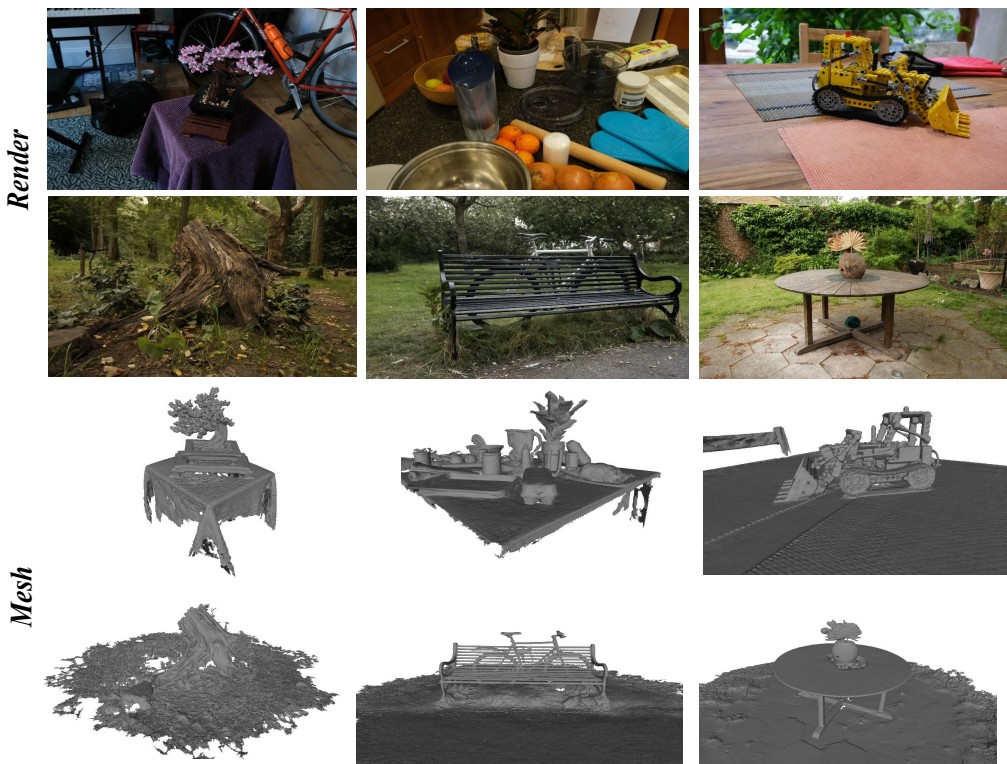

Figure F: **Qualitative results on the Mip-NeRF360 dataset.** Our method reconstructs surfaces with fine geometry details and produces high-fidelity renderings on Mip-NeRF360 dataset.

### A.8 BROADER SOCIAL IMPACTS

This work presents Hierarchical Gaussian Splatting (HRGS), a super-resolution reconstruction framework tailored for computationally constrained environments. HRGS achieves high-fidelity 3D reconstruction while significantly reducing GPU memory usage and model size, thereby enhancing the accessibility of advanced 3D vision techniques. Its efficiency enables deployment on low-power platforms such as mobile and embedded devices, with promising applications in education, cultural heritage preservation, smart city visualization, and immersive virtual or augmented reality. However, we acknowledge the potential for misuse in privacy-sensitive contexts, such as unauthorized spatial reconstruction. To mitigate such risks, the deployment of HRGS should be governed by clear ethical guidelines and regulatory oversight.

## B USE OF LARGE LANGUAGE MODELS

A large language model (LLM) was used solely for language-level assistance, such as improving readability, fluency of the text and formatting LaTeX tables and retrieve related works. The research ideas, experiments, and results are entirely the work of the authors, who bear full responsibility for the content of this submission.

