# OpenReview forum: "HRGS: Hierarchical Gaussian Splatting for Memory-Efficient High-Resolution 3D Reconstruction"
_ICLR.cc/2026/Conference — ICLR 2026 Conference Withdrawn Submission_

### Official Review · Reviewer_Ef38 · 2025-10-23

**Soundness:** 2
**Presentation:** 3
**Contribution:** 2
**Rating:** 2
**Confidence:** 4

**Summary:**

This paper proposes Hierarchical Gaussian Splatting (HRGS), a coarse-to-fine framework designed to achieve memory-efficient, high-resolution 3D reconstruction. The method introduces (1) hierarchical block-wise optimization using low-resolution global Gaussians as priors, (2) a data and Gaussian partitioning strategy combining SSIM-based assignment and spatial contraction, and (3) an Importance-Driven Gaussian Pruning (IDGP) mechanism to remove low-impact primitives.

**Strengths:**

1. Addresses a relevant problem: scaling 3D Gaussian Splatting to high-resolution scenes under limited GPU memory.

2. Integrates known concepts (hierarchical refinement, pruning, data partitioning) into a unified training pipeline.

3. Provides extensive comparisons across multiple datasets with standard metrics.

**Weaknesses:**

1. Misaligned comparison with Mip-Splatting: The paper repeatedly compares HRGS against Mip-Splatting (Yu et al., 2024) and even highlights marginal improvements in PSNR and SSIM. However, this comparison is conceptually inappropriate: Mip-Splatting is primarily designed for alias-free radiance rendering, not for memory-efficient or block-wise optimization.

2. Incomplete ablation and missing baselines: Although the appendix mentions additional ablation studies, the main text provides only brief tables without qualitative results. The paper also omits comparisons with several directly relevant hierarchical methods, making it difficult to contextualize the claimed advantages.

3. Limited analysis and theoretical support: The paper provides no theoretical rationale for why hierarchical block optimization improves convergence or scalability. The choice of SSIM threshold, pruning ratio, and number of blocks are all heuristic and dataset-dependent.

**Questions:**

1. How sensitive is the method to the number of blocks and pruning ratio? Is there an optimal trade-off between memory efficiency and reconstruction quality?

2. How does HRGS compare to Octress-GS, which also uses hierarchical Gaussian partitioning? Why is it not included in the comparisons?

3. The method introduces multiple heuristics. Are these hyperparameters fixed across datasets, and how robust are the results to their variation?

---

### Official Review · Reviewer_ds5p · 2025-10-26

**Soundness:** 3
**Presentation:** 2
**Contribution:** 2
**Rating:** 4
**Confidence:** 4

**Summary:**

The authors propose HRGS, a method to scale up volumetric and surface reconstruction quality over high resolution images. HRGS takes a divide and conquer approach: it first reconstruct the scene coarsely to define the object space, then perform warping/contraction on the coordinate field, based on pre-defined regions, such that gaussians can be more uniformly divided. The divided Gaussian assignments are chosen based on several metrics, e.g. contribution to SSIM and whether the cameras lie in the block. HRGS further introduces Importance Driven Gaussian Pruning, which applies lightweight scoring and pruning strategy by removing lowest 20% of guassians with lower contribution. The loss function adds additional constrains on depth and normal, which are implemented from VCR-Gaus.

**Strengths:**

The proposed concept is sound, and addresses a critical issue of scaling 3DGS when it's nontrivial to breakup the gaussians properly.

The overall approach shows some metric improvement in NVS and surface reconstruction. The authors show that this is done with a balance of producing high resolution detail and low memory footprint on standard benchmark.

**Weaknesses:**

I think this work, while has good motivation, suffers from trying to achieve multiple things at the same time, with confusing message in the end. It is unclear to me if the authors want to focus on e.g., efficient GS that leads to HR rendering, or surface reconstruction. I would be interested to see, e.g., just by scaling GS subdivision, how far can we get to the best performance possible. In my mind, if I have infinite memory, I should be able to scale GS to densify as much as it needs to, and stabilize. Based on the ablation study, if HRGS splits to too many subblocks, the rendering quality quickly degrades.

The contributions are on the thinner side, even though I think the topic of scaling GS to high resolution is very important.

I wonder why the authors only tested two datasets, T&T and MipNerf360, one for surface reconstruction and one for NVS. There are plenty of high resolution datasets, and the authors should be able to collect more with just consumer phones. I think one dataset for one task is on the lower end and does not prove generality yet.

For efficient GS and surface reconstruction GS, the below literature is a snippet of what should be considered and compared:
Efficient GS:
EAGLES (Girish et al.)
PUP 3D-GS (Hanson et al.)
LightGaussian (Fan et al.)
LP-3DGS (Zhang et al.)

Surface reconstruction:
RaDeGS (Zhang et al.)
Gaussian Opacity Fields (Yu et al.)
Geometry Splatting (Jiang et al.)

**Questions:**

An important question: How does this work for non object-centric scenes, where the definition of center is not clear? E.g., for Autonomous Vehicle data.

What is the primitive used for HRGS? 2D or 3D gaussians?

Can authors explain why too many subdivisions lead to such strong overfitting effects? Seems to me to be a very big issue as this implies HRGS is not very generalizable.

---

### Official Review · Reviewer_qAD4 · 2025-11-01

**Soundness:** 3
**Presentation:** 3
**Contribution:** 3
**Rating:** 4
**Confidence:** 4

**Summary:**

This paper introduces HRGS, a method to solve the high memory usage problem of 3D Gaussian Splatting (3DGS) when reconstructing high-resolution 3D scenes. Standard 3DGS fails on GPUs with limited memory for 4K-5K resolution images. HRGS uses a two-step, "coarse-to-fine" approach: first, it creates a low-resolution, coarse model of the entire scene. Then, it splits the scene into smaller blocks and refines each one separately using high-resolution images, guided by the initial coarse model to ensure the blocks fit together seamlessly. It also uses a smart pruning technique to remove unnecessary 3D Gaussians during training, saving more memory. This allows HRGS to achieve high-quality, high-resolution 3D reconstruction and novel view synthesis on standard 24GB GPUs, outperforming other methods in quality while using significantly less memory and storage.

**Strengths:**

-  Directly and effectively addresses the major scalability issue of 3DGS, enabling high-resolution (5K) reconstruction on commodity, memory-constrained GPUs where other methods fail.
- The paper proposes an effective contrastive method to ensure that the Gaussians are properly separated, providing a more reasonable and principled paradigm.
- The performance gains are clear and substantial. HRGS achieves superior PSNR and SSIM on the Mip-NeRF360 dataset while using nearly half the GPU memory of 3DGS and Mip-Splatting and producing significantly smaller models.

**Weaknesses:**

- The two-stage, block-wise process is inherently more complex and slower than a single global optimization. The paper reports a training time of 5 hours for HRGS vs. 3 hours for 3DGS, representing a trade-off of time for memory efficiency. I also suggest reporting comparisons using the same GPU type and normalized GPU hours to ensure fairness in evaluating efficiency.
- The proposed Importance-Driven Gaussian Pruning (IDGP) induces resolution-dependent information loss, disproportionately removing Gaussians that encode high-frequency structures. When resolution increases, these structures disappear or attenuate according to Figure D (second row).
- The framework introduces several new hyperparameters and algorithmic steps, such as the number of blocks, SSIM threshold for data selection, and the IDGP pruning rate. This increased complexity could pose a challenge for reproduction and tuning for new datasets.

**Questions:**

Please refer to the weaknesses listed above.

---

### Official Review · Reviewer_2fNj · 2025-11-01

**Soundness:** 3
**Presentation:** 2
**Contribution:** 2
**Rating:** 4
**Confidence:** 4

**Summary:**

To enable high-resolution novel-view synthesis on consumer-grade GPUs (e.g. with 24 GB of VRAM), HRGS offers a memory-efficient 3D GS optimization framework that relies on coarse-to-fine hierarchical block-level parallel optimization. A set of 3D GS initialization points in a scene is first re-mapped to a contracted space, then uniformly divided into 3D blocks in this contracted space. Then, relevant training camera poses (which are not necessarily disjoint) are assigned to each block based on the locations of the camera poses and how much they contribute to the initial coarse-Gaussians rendering of each block, measured via a SSIM loss. Then, every block is initialized with the global coarse Gaussians it encloses and optimized independently with reduced number of Gaussians and training views. As a result, HRGS reduces the peak VRAM usage, by about 39% compared to vanilla 3D GS and 54% from Mip-Splatting.

**Strengths:**

HRGS is much faster and more memory-efficient compared to competing methods for novel-view synthesis and surface reconstruction (Table 1-3, Figure C, Table F). At the same time, it achieves state-of-the-art image quality and F1 scores on the Mip-NeRF 360 and Replica datasets, respectively.

Being able to do high-resolution novel-view synthesis in a memory-efficient way is a very impactful research area, in my opinion. Consumer GPUs (e.g. with 24 GB of VRAM) frequently do not have enough memory to naively do high-resolution NVS, even though they're capable in terms of computing power.

**Weaknesses:**

I am not fully convinced that the contraction algorithm will work well for scenes that are larger or sparser in camera views (e.g. a long, curved corridor). The contraction algorithm assumes an "internal region" of the scene bounding box to be at the center (L245), but the center of the scene may be empty for some scenes (e.g. a circular path around a building external). Additionally, I belief the nonlinear mapping rule only applies a scalar scaling (Eq. 2) of points from the origin, which again may not be flexible enough for diverse scenes. I'm very curious to see how well HRGS works on larger-scale, more diverse scenes, such as on a random subset of scenes from DL3DV. Larger scenes may have a great horizontal scale but small vertical scale, which challenges the uniformity offered through the scalar contraction mapping in Eq. 2.

I feel that comparisons to 3DGS, 3DGS+EWA, Mip-Splatting in the experiments (e.g. Table 1, Table 2) are not exactly fair. HRGS incorporates depth and normal priors from pretrained vision models while the other methods do not. On the other hand, 3DGS, 3DGS+EWA, Mip-Splatting can also incorporate these depth and normal priors to likely improve quality. The confounding factor of incorporating depth and normal priors makes it hard to disentangle the paper's main contribution, which is about coarse to fine hierarchical block-level optimization rather than the incorporation of depth and normal priors. I'm curious to see additional evals where this confounding factor is eliminated.

A minor point, the paper leaves some important details unexplained, such as how rendered normal map \hat{N} (L305, Eq. 7) are obtained, and what L_{s} is (L323, Eq. 10). I think it makes it hard for the average reader to grasp the paper's method. I expect greatly improved clarity in Section 3: Methodology. Another minor point, but the paper needs more careful polishing in general, e.g. as the missing mention of units/metrics in Table 2. Not a major weakness because I expect these will be fixed in a future version, but it does make reading the paper at its current state more difficult than it should be.

**Questions:**

In Fig 1 (b) and (c), over what datasets are these averaged?

In L305, how is the rendered normal map \hat{N} obtained from the HRGS model? It is never explained in the paper.

In L319, how is the confidence term "w" obtained? What does this intuitively represent and why do we need it?

In L323, I think L_s should be briefly described instead of left as a reference to another paper, as it is an important part of the loss.

What are the units/metrics for the quantitative results in Table 2? The text says it's F1 but the table does not make this clear.

Figure C should probably just be a bar chart? The lines drawn between the different VRAM usages of different methods is confusing, as these are simply different methods.

Minor Points:
In Fig 1 (b), why is VRAM usage of different methods connected with a line, while other statistics are not? PSNR is a bar, while others are points. This chart is a bit confusingly formatted.

---

### Note · Authors · 2026-01-04

I have read and agree with the venue's withdrawal policy on behalf of myself and my co-authors.